# Impact of COVID-19 on the Environments of Professional Nursing Practice and Nurses’ Job Satisfaction

**DOI:** 10.3390/ijerph192416908

**Published:** 2022-12-16

**Authors:** Olga Maria Pimenta Lopes Ribeiro, Vânia Maria Oliveira Coimbra, Soraia Cristina de Abreu Pereira, Ana da Conceição Alves Faria, Paulo João Figueiredo Cabral Teles, Carla Gomes da Rocha

**Affiliations:** 1Nursing School of Porto (ESEP), 4200-072 Porto, Portugal; 2CINTESIS@RISE, 4050-313 Porto, Portugal; 3North Region Health Administration, 4000-447 Porto, Portugal; 4School of Economics, University of Porto, 4200-465 Porto, Portugal; 5Laboratory of Artificial Intelligence and Decision Support—INESC Porto LA, 4200-465 Porto, Portugal; 6Institute of Health, School of Health Sciences, HES-SO Valais-Wallis, 1950 Sion, Switzerland

**Keywords:** coronavirus infection, hospitals, job satisfaction, pandemic, work environment

## Abstract

(1) Background: The repercussions of work environments were widely studied before the pandemic. However, there are still many difficulties to be discovered considering the impact generated by it. Thus, this study aimed to analyse the impact of COVID-19 on nursing practice environments and nurses’ job satisfaction. (2) Methods: A correlational study was conducted in a hospital in northern Portugal, with the participation of 416 registered nurses. Data were collected in June 2021 through questionnaires. The study was approved by the Institutional Ethics Committee. (3) Results: COVID-19 had a favourable impact on the structure component of the practice environments; the process component decreased compared to the pre-pandemic period; the outcome component remained moderately favourable to the quality of care. Nurses were not very satisfied or not at all satisfied with their valuation and remuneration; moderately satisfied with the leadership and staffing; and satisfied with the organisation and resources, co-workers and valuation by patients and families. In more favourable environments, nurses’ job satisfactions were higher. (4) Conclusions: Identifying the dimensions with the best and worst scores allowed the institution’s managers to concentrate efforts on where improvements were needed, thus preparing professional contexts for the recovery of care activities.

## 1. Introduction

In the last few decades, the increasingly complex care needs, technological evolution, and the concern with guaranteeing the quality of care have constituted an enormous challenge for nursing [1]. However, due to the fragilities inherent to the work contexts, violence, burnout, professional dissatisfaction, and the intention to leave the workplace and/or the profession have worsened in recent years [2].

In addition, since December 2019, the world has been facing a COVID-19 outbreak. This disease, caused by the SARS-CoV-2, which due to its rapid global spread was classified as a pandemic in March 2020 by the World Health Organization (WHO) [3].

The sudden onset of a potentially fatal disease, by exponentially increasing care needs, placed an extraordinary pressure on all health systems, requiring significant changes and adjustments in practice settings [4], both in hospitals and in primary health care institutions. In fact, in addition to treating patients with COVID-19 with the need for hospitalisation, at a territorial level, primary health care professionals stood out in the planning and implementation of preventive strategies, fundamental to the control of the pandemic [5]. Without this work in the preventive field, the pressure on the health systems of different countries and, consequently, on nursing professionals, would have been even more significant.

The workload, available resources, participation in decisions, interpersonal relationships, feedback, and support are factors that have an impact on nursing professionals, having consequences not only on their well-being, but also on their performance [1,6]. Feelings of satisfaction and dissatisfaction occur in response to internal and external experiences in the work environment, and although they may be associated with personal characteristics of the professionals, they are highly related to working conditions [1,7].

In a pandemic context, the daily work problems of nursing professionals have been described in the international literature, including poor working conditions, lack of personal protective equipment, lack of professionals, increased workload, physical and psychological overload, as well as low remuneration [8]. Therefore, the pandemic seems to have a negative influence on job satisfaction [9]. This is worrisome, since job satisfaction is a critical motivational resource to avoid burnout, which in the context of a pandemic is even more relevant [10].

Results from a study conducted in Quebec, with the participation of 1705 nurses, showed that nursing staff caring for patients with COVID-19 perceived less transformational leadership from their superior, had higher chronic fatigue, lower job satisfaction, lower perceived quality of care, and higher intention to leave their current professional position. Job demands (caring for patients with COVID-19), job resources (transformational leadership, knowledge/preparedness), as well as strain indicators (chronic fatigue and job satisfaction) were found to be significant determinants of nursing staff turnover intentions [11]. In this context, higher chronic fatigue, perceived poorer quality of care, lower job satisfaction, and higher intention to leave the organisation were identified in nurses caring for patients with COVID-19 [11].

In a study conducted during the first critical period of the pandemic, with the participation of 767 Portuguese nurses [12], the authors concluded that these professionals had higher levels of depression, anxiety and stress than the general population, which made it necessary for several institutions to define and implement strategies to minimize the negative repercussions of the pandemic on these professionals. In fact, a good management of psychosocial risks, associated with job satisfaction, is essential and may act as a protective factor against the risks associated with work stress.

Even before the pandemic, the repercussions of the professional practice environments on nurses, clients, and institutions had been widely studied at an international level. The pandemic exacerbated some of the difficulties that nurses were already facing, particularly in hospital settings. Although the situation of COVID-19 triggered a series of responses to address some of the greatest emerging problems [6], in view of the impact generated by the pandemic there are still many difficulties to overcome.

In this sense, in the different work contexts, knowing the characteristics of the professional practice environments and the nurses’ levels of job satisfaction will make it possible to (re)think the strategies, in order to guarantee practice environments that promote the quality of care, and the well-being of the professionals.

Thus, due to the need to understand the impact of the pandemic on work environments, this study aimed to analyse the impact of COVID-19 on nursing practice environments, and on nurses’ job satisfaction.

## 2. Materials and Methods

This is a quantitative, correlational study guided by the Strengthening the Reporting of Observational Studies in Epidemiology (STROBE^®^) tool.

### 2.1. Setting and Participants

The study was conducted at a university hospital in northern Portugal, which, since the beginning of the pandemic, was considered a reference hospital for the care of patients with COVID-19; it stood out by responding to the population’s needs and anticipating the problems that emerged in the various critical periods of the pandemic in this country [13].

The inclusion criteria for the selection of participants were defined as follows: nurse or specialist nurse working in the adult inpatient services of the departments of Medicine, Surgery, or Emergency and Intensive Care; and having worked in the institution for more than 18 months, i.e., having worked in the institution since the pre-pandemic period. All professionals absent due to leave during the data collection period were excluded. Thus, from a population of 713 nurses working in the departments mentioned above, using a non-probability convenience sampling technique, 416 registered nurses participated in the study, which corresponds to 58.3% of the population.

### 2.2. Instruments

A self-completion questionnaire consisting of three parts was used for data collection: the first part included the participants’ sociodemographic and professional characterisation; the second part included the Scale for the Environments Evaluation of Professional Nursing Practice (SEE-Nursing Practice) [14]; the third part included the Nurse Job Satisfaction Scale [15].

The SEE-Nursing Practice is composed of three subscales: the SEE-Nursing Practice—Structure, composed of 43 items divided into six dimensions; the SEE-Nursing Practice—Process, composed of 37 items divided into six dimensions; and the SEE-Nursing Practice—Outcome, with 13 items divided into two dimensions. In the three subscales, each item was answered on a Likert-type scale with five options, where one corresponds to “never”, two “rarely”, three “sometimes”, four “often”, and five “always”.

The Nurse Job Satisfaction Scale is composed of 37 items distributed by 6 dimensions. Each item is answered on a Likert scale with 5 options, where “1” corresponds to “absolutely nothing”, “2” to “a little”, “3” to “moderately”, “4” to “very”, and “5” to “extremely”.

### 2.3. Data Collection and Procedures

Data were collected through two researchers who visited each department taking part in the study to deliver the questionnaires, which were subsequently collected in the same place, based on prior scheduling and availability of the professionals involved. In addition to the written information, which was attached to the questionnaire, the research was also presented in person to the nurses. After the objective’s clarification, the nurses were free to fill in—or not fill in—the questionnaire, subsequently placing it in a closed envelope. It should be noted that incomplete questionnaires were rejected, thus not being considered for analysis.

When completing the questionnaire, regarding the items of the SEE-Nursing Practice [14] and Nurse Job Satisfaction Scale [15], participants were asked to answer in relation to two distinct moments in time: before the pandemic, and the present moment, which, in this study, corresponded to the 3rd critical period of the COVID-19 pandemic in Portugal. The critical period was considered as the one in which there was a higher number of patients hospitalised due to COVID-19, with a subsequent decrease in the number of new cases and deaths. Consequently, data collection took place from 1 June to 30 June 2021. Note that although the month of data collection corresponded to the recovery time of the 3rd critical period, a 4th critical period was already imminent in the country, which occurred in July and August 2021 [16].

### 2.4. Data Analysis

Descriptive and inferential statistics were performed using the Statistical Package for the Social Sciences (SPSS) software, version 26.0. When analysing the results, the higher the score of the SEE-Nursing Practice, the more favourable the environment of professional nursing practice is to the quality of care. Regarding the subscales, the higher the score, the more the structure, process or outcome are favourable to the quality of care. With regards to the Nurse Job Satisfaction Scale, the higher the score, the higher the nurses’ satisfaction with their work.

To analyse the results related to the environments of professional nursing practice, the following criteria were established: score <35%—component of the environment of nursing professional practice not very favourable to the quality of care; from 35% to 55%—component of the environment of nursing professional practice moderately favourable to the quality of care; from 55% to 75%—component of the environment of professional nursing practice favourable to the quality of care; and, finally, >75%—component of the environment of nursing professional practice very favourable to the quality of care.

Regarding nurses’ job satisfaction, it was determined that score <35%—not very satisfied; from 35% to 55%—moderately satisfied; from 55% to 75%—satisfied; and finally >75%—very satisfied.

At the beginning of the statistical analysis, using the Shapiro–Wilk and Lilliefors tests, the normality was rejected for all dimensions and scales/subscales. Therefore, for the variables “environments of professional nursing practice” and “nurses’ job satisfaction”, the comparisons between the pre-pandemic period and the period after the 3rd critical period of COVID-19 were carried out using the Wilcoxon’s test for paired samples. The significance level adopted was 0.05. Subsequently, to test the relationships between the environments of professional nursing practice and the nurses’ job satisfaction, Spearman’s correlation coefficient was used.

### 2.5. Ethical Considerations

The study was approved by the Ethics Committee of the hospital under No. 104/21. All those who agreed to participate signed the informed consent form. Confidentiality and anonymity were guaranteed in the use and disclosure of the obtained data.

## 3. Results

A total of 416 nurses participated in the study. Of these participants, 264 (63.5%) practiced as registered nurses, and 152 (36.5%) as specialist nurses. Regarding the area of specialisation, 59 (38.8%) were specialists in medical-surgical nursing, 71 (46.7%) in rehabilitation nursing, 8 (5.3%) in community health, 6 (3.9%) in community public health, and the remaining 8 (5.3%) specialised in other areas. Their sociodemographic and professional characteristics can be found in Table 1.

In relation to the professional nursing practice environments, the results are presented in Figure 1, Figure 2 and Figure 3.

In all dimensions of the component structure and in the subscale itself, the mean percentage was higher after the 3rd critical period of COVID-19 in Portugal. Regarding the process, the mean was the same in both periods for the dimensions “collaboration and teamwork” and “theoretical and legal support of professional practice.” The dimensions “autonomous practices in professional practice”, “care planning, evaluation and continuity”, and “interdependent practices in professional practice” presented a lower mean after the 3rd critical period of COVID-19. In the dimension “strategies for ensuring quality in professional practice”, the mean was slightly higher after the 3rd critical COVID-19 period, when compared to the pre-pandemic period. About the outcome, the mean was higher after the 3rd critical period of COVID-19 for both dimensions.

Regarding nurses’ job satisfaction, the results are shown in Table 2.

In the dimensions “satisfaction with the leadership”, “satisfaction with professional recognition”, and “satisfaction with the co-workers”, the mean was higher after the 3rd critical period of COVID-19. In the dimensions “satisfaction with recognition and remuneration”, and “satisfaction with staffing”, the mean was lower after the 3rd critical period of COVID-19. The “satisfaction with the organisation and resources” maintained the same mean in both periods.

When the relationship between the components of professional practice environments and nurses’ job satisfaction was analysed, all correlations were significant and positive; that is, a higher score of each component of the practice environment corresponded to a higher score of any dimension of satisfaction, according to Table 3.

Regarding the structure component, in the pre-pandemic period, correlations with the dimensions “satisfaction with the co-workers”, “satisfaction with recognition and remuneration”, and “satisfaction with staffing” were weak, while the correlations with the dimensions “satisfaction with the leadership”, “satisfaction with the organisation and resources”, “satisfaction with professional recognition”, and with the global scale, were moderate. After the 3rd critical period of COVID-19, the number of moderate correlations increased. At this stage, there was a weak correlation with the dimension “satisfaction with recognition and remuneration”, moderate correlations with the dimensions “satisfaction with the leadership”, “satisfaction with the organisation and resources”, “satisfaction with professional recognition”, “satisfaction with the co-workers”, and “satisfaction with staffing”, and a strong correlation with the global scale.

Regarding the process component, in the pre-pandemic period, the correlations with the dimensions “satisfaction with professional recognition”, “satisfaction with recognition and remuneration”, and “satisfaction with staffing” were weak, while the correlations with the dimensions “satisfaction with the leadership”, “satisfaction with the organisation and resources”, “satisfaction with the co-workers”, and with the global scale were moderate. After the 3rd critical period of COVID-19, the exact same results were found.

Regarding the outcome component, in the pre-pandemic period, the correlations with the dimensions “satisfaction with professional recognition” and “satisfaction with staffing” were weak, while the correlations with the dimensions “satisfaction with the leadership”, “satisfaction with the organisation and resources”, “satisfaction with the co-workers”, “satisfaction with recognition and remuneration”, and with the global scale were moderate. After the 3rd critical period of COVID-19, correlations were found to be weak with the dimension “satisfaction with recognition and remuneration”, and moderate with the dimensions “satisfaction with the leadership”, “satisfaction with the organisation and resources”, “satisfaction with professional recognition”, “satisfaction with the co-workers”, “satisfaction with staffing”, and with the global scale.

## 4. Discussion

According to the Order of Nurses of Portugal, the professional association that regulates the nursing profession in Portugal, in December 2020, 82.3% of the nurses working in the country were women and 17.7% were men. The age group 36 to 40 years old was the most prevalent [17]. In this study, the percentage of male nurses was higher (25.5%), while the mean age (38.3 ± 8.5) was concordant. As for the academic degree and field of specialisation, the predominance of bachelor’s degree and specialisation in rehabilitation nursing reflects the national data [17].

Regarding the environments of professional nursing practice, in the component structure, a positive trend was confirmed in all dimensions, with higher mean scores after the 3rd critical COVID-19 period. The fact that the COVID-19 outbreak required quick adaptation led institutions to invest in creating conditions that allowed for better care during the pandemic, leading to an improvement of this component in the environments of practice [4,6,18].

In fact, in the hospital in question, the rapid adaptation to the needs of the community and professionals, and to the epidemiological profile of the cases was essential. The investment was made to increase the availability of material resources, the hiring of more nursing professionals, the mobilisation of nurses from services which reduced their activity to those which ensured the care of patients with COVID-19, and the strict separation of COVID-19 and non-COVID-19 areas [13]. Overall, this had a positive impact on the structure component.

Both in the pre-pandemic context and after the 3rd critical period of COVID-19, the dimensions with the worst percentages were “institutional policy for professional qualification” and “nurses’ participation and involvement in the institution’s policies, strategies and management”. The little participation in the decisions of the institution and the negative repercussion in the work environments was also pointed out by nurses from three university hospitals in Brazil [19]. Although the demand for a quick response has enhanced the centralisation of decision making in institutional managers [19], the results once again reinforce the need to provide conditions for the participation of nurses, as well as their professional qualification.

The involvement of the nurses in decision-making, in addition to giving them the opportunity to express personal views, increases the sense of belonging to the institution, mutual esteem, and teamwork; this has a positive repercussion on job satisfaction and, subsequently, on the improvement of health quality indicators [20].

Although the process component is the one with the best global score, when compared to the pre-pandemic phase, it showed a decrease in the percentage of “autonomous practices in professional practice”, “care planning, evaluation and continuity”, and “interdependent practices in professional practice” after the 3rd critical period of COVID-19. These results presuppose that the fact that nurses have worked for more than a year in unfavourable contexts and with heavy workloads, has led to negative repercussions on their professional practice, both in the autonomous and interdependent domains of the profession. In relation to the dimension “strategies for ensuring quality in professional practice”, a positive trend was found.

Following the concern to ensure the quality of the provided care, in the outcome component, the “systematic assessment of nursing care and indicators” and the “systematic assessment of nurses’ performance and supervision” were also better scored after the 3rd critical period of COVID-19, denoting a growing concern to demonstrate the results achieved in fighting the pandemic [4].

The fact that the hospital in question had become a national reference for the care of patients with COVID-19 [13], also led to a greater focus on the assessment of the strategies/measures that were being implemented, which consequently had a positive impact on the outcome component.

Regarding professional satisfaction, in the dimensions “satisfaction with the leadership”, “satisfaction with professional recognition”, and “satisfaction with the co-workers”, the mean was higher after the 3rd critical period of COVID-19. Facing a real public health crisis, health professionals, namely nurses, not only had to work longer hours, but often did so in contexts not favourable to the quality of care, or to their own well-being [21,22]. In this sense, it is crucial that they are and feel supported. Adequate management, leadership, and team spirit can create favourable working conditions and reduce physical fatigue and psychological stress, strengthening the resilience of those who will have to remain on the front line [21]. The pandemic dictated the reinforcement of teams and required increased surveillance from nurse managers to ensure their management, both in terms of availability and skills, in order to optimize the efficiency and productivity of teams, guaranteeing their safety, and the best possible care to patients [6,23].

In a study conducted in Brazil, the authors found that when faced with the pandemic crisis scenario, nurse managers played a key role, not only due to their fundamental role in the provision of physical, human, and material resources necessary to ensure safe and quality care, but also in supporting nurse care providers—not only in the scientific and instrumental domains, but also emotionally [24], an aspect which has a direct impact on the professionals’ satisfaction with their leadership.

In a systematic literature review, the authors concluded that nurse leaders are indispensable for creating positive work environments that retain an empowered and motivated workforce. In studies included in this review, it is concluded that positive and supportive leadership styles can improve job satisfaction, organisational commitment, and nurses’ intentions to remain in the unit, institution, and profession, while reducing emotional exhaustion [20].

The trend of the dimensions “satisfaction with staffing” and “satisfaction with recognition and remuneration” was negative. In all dimensions, the scores ranged from “moderately satisfied” to “satisfied”—except for the dimension “satisfaction with recognition and remuneration”, in which most nurses stated to be “little” or “not at all” satisfied. A study carried out in Israel, during the pandemic, found that the mean of job satisfaction also ranged from moderate to high, showing that despite the difficult conditions, nurses are highly motivated professionals who strongly adhere to the social demands of the profession [25]. Although it was expected that the COVID-19 pandemic would have a negative repercussion on professional satisfaction [9], the results of this study show a slight increase in the mean percentage. The fact that the research was not conducted during all stages of the pandemic limits the understanding of the repercussions on job satisfaction. In a study carried out in five hospitals in the Philippines, an increased level of fear of COVID-19 was associated with decreased job satisfaction, increased psychological distress, and increased institutional and professional turnover intentions. In the same study, the authors concluded that dealing with the fear of COVID-19 may lead to better outcomes for frontline nurses, increasing their job satisfaction, reducing their stress levels, and their intentions to leave the institution and profession [26]. Considering that this study evaluated nurses’ job satisfaction at the pre-pandemic moment and after the 3rd critical period of COVID-19, which occurred one year after it started, the findings regarding nurses’ satisfaction may be related to the adoption of strategies to mitigate fear, and its repercussions. In addition, it is possible that the nurses had adapted to the “new normal”, which accompanied by the investment of the institutions’ management bodies, culminated in improved job satisfaction.

The significant and positive correlations, between the variables studied, prove that positive professional practice environments will contribute to improve nurses’ job satisfaction, which had, in fact, already been confirmed in studies carried out before the pandemic [2,27,28].

It should be highlighted that after the 3rd COVID-19 critical period, the correlations between the three components of practice environment and job satisfaction are weak in relation to the dimension “satisfaction with recognition and remuneration”. In fact, although in many situations, as was the case in the institution under study, strategies were adopted to ensure the quality of care and the nurses’ well-being, the worst-scoring items refer to the satisfaction with the salary considering the skills/knowledge, and the satisfaction with the salary considering the executed tasks, which is not effectively under the responsibility and autonomy of public hospital institutions.

Although it is not always possible to change some less favourable aspects, nurse managers can make a difference by supporting the workforce and creating safe and healthy environments where the team can give their best, thus contributing to the nurses’ well-being, and greater commitment to the profession and institution [1,27]. Formal and informal recognition can enhance positive feelings in the work environment and can extend to the whole team [1].

Studies have revealed that transformational leadership has a significant positive correlation with nursing job satisfaction levels. This means that if nursing managers are transformational leaders, through their inspirational and motivational behaviour, in addition to contributing to building positive environments, they can induce changes in the psychological states of nurses working within the organisations [20]. In this context, during COVID-19, the primary role of managers in promoting the mental health and psychological resilience of professionals was evidenced through the provision of assistance mechanisms and psychological support [29].

Focusing on the impact of COVID-19 on practice environments and nurses’ job satisfaction, this study provided important information to the extent that it showed that even in periods of crisis, the investment in the various dimensions of work environments has a favourable impact on nurses’ job satisfaction. However, considering that the rapid evolution of SARS-Cov2 did not allow for preparation within the different contexts of professional practice, during the time of recovery from the pandemic, investment in work environments became more important than ever [1]. In this context, the investment in transformational and authentic nursing leadership [20], the investment in professional qualification, the nurses’ participation in the decision-making process of the institution [19], as well as the appreciation and recognition of their role, will be decisive for professional satisfaction and, more broadly, for the creation of more positive environments for professionals, patients, and institutions themselves.

In people management, it is necessary for nurse managers to keep in mind the basic functions of management: motivation, planning, and organisation. With regard to motivation, it should be noted that an adequately motivated professional is key in achieving the organisation’s objectives, since motivation promotes satisfaction, involvement, loyalty and the willingness to do more and do it better [30]. Thus, it is essential to invest in nurses’ motivation.

Despite the relevance of the results, this study has some limitations. First, during data collection, participants were asked to answer in relation to two different moments in time: before the pandemic, and at the current moment (3rd critical period of the COVID-19 pandemic in Portugal). Although this data collection strategy allowed us to assess the impact of the pandemic on the dimensions of practice environments and job satisfaction, we assumed the risk of response bias. Second, was the fact that this research was conducted in a single institution. Thus, studies in other institutions are suggested, in order to contribute to the validation and generalisation of the results. However, by providing an initial understanding of the variables studied in a pandemic context, the results are extremely relevant for nursing and for the institution managers, to the extent that they allow rethinking of strategies to improve work environments, with repercussions on professional satisfaction.

## 5. Conclusions

The results of this study confirmed that the structure component of the environments of professional nursing practice, within the institution studied, went from “moderately favourable” to “favourable” regarding the quality of care; the process component, although favourable to the quality of care in both moments, registered a lower percentage value after the 3rd critical period of COVID-19; in the outcome subscale, although a positive trend was recorded, the mean percentage indicates that this component continues to be moderately favourable to the quality of care.

Regarding professional satisfaction, nurses remain poorly satisfied with their recognition and remuneration; they remain moderately satisfied with supervisors and staffing; and they expressed that they were satisfied with the organisation and resources, their co-workers, and with the professional recognition by patients and their families.

An environment of professional practice favourable to the quality of care is significantly correlated with nurses’ job satisfaction. In this sense, in addition to evaluation, the definition of strategies that make nursing practice environments more favourable will have important repercussions on nurses’ job satisfaction, which will boost their ability to continuously ensure the quality of their provided care.

## Figures and Tables

**Figure 1 ijerph-19-16908-f001:**
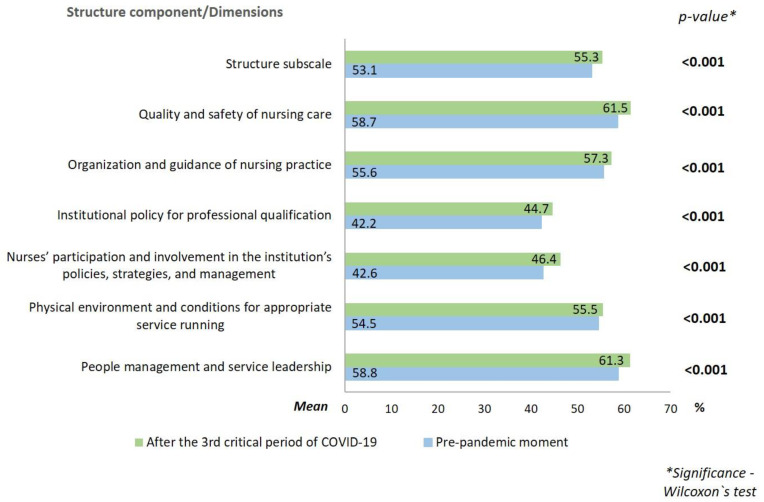
Mean scores of the structure component of nursing professional practice environments at the pre-pandemic moment and after the 3rd critical period of COVID-19.

**Figure 2 ijerph-19-16908-f002:**
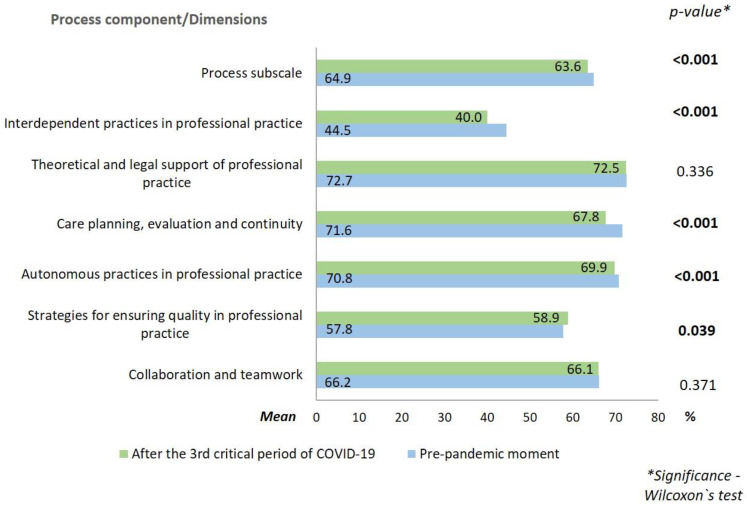
Mean scores of the process component of nursing professional practice environments at the pre-pandemic moment and after the 3rd critical period of COVID-19.

**Figure 3 ijerph-19-16908-f003:**
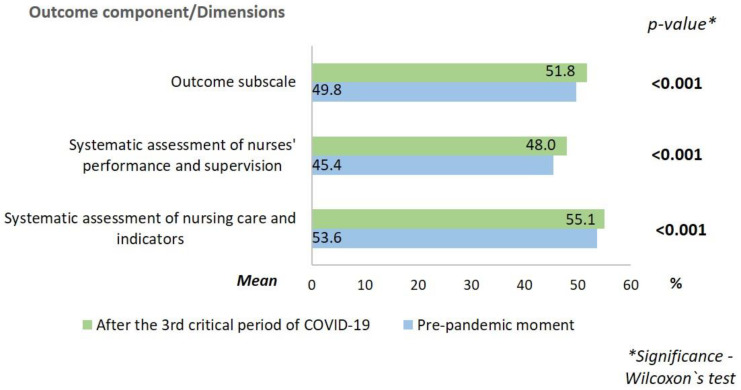
Mean scores of the outcome component of nursing professional practice environments at the pre-pandemic moment and after the 3rd critical period of COVID-19.

**Table 1 ijerph-19-16908-t001:** Sociodemographic and professional characterisation of the participants (*n* = 416).

Gender *n* (%)	
Male	106 (25.5)
Female	310 (74.5)
Marital status *n* (%)	
Single	148 (35.6)
Married/Non-marital partnership	248 (59.6)
Divorced	19 (4.6)
Widowed	1 (0.2)
Age (years) Mean; Median; Std. Dev. *	38.3; 37; 8.5
Education *n* (%)	
Bachelor’s degree	359 (86.3)
Master’s degree	57 (13.7)
Work Department *n* (%)	
Medicine Department	176 (42.3)
Surgery Department	141 (33.9)
Emergency and Intensive Care Department	99 (23.8)
Areas of care for COVID-19 patients	279 (67.1)
Time (months) Mean; Median; Std. Dev. *	7.7; 7; 4.2
Professional category *n* (%)	
Nurse	264 (63.5)
Specialist nurse	152 (36.5)
Time of professional practice (years) Mean; Median; Std. Dev.	15; 14; 8.5
Time of professional practice in the service (years) Mean; Median; Std. Dev. *	9.8; 9; 8.1

* Std. Dev.—Standard Deviation.

**Table 2 ijerph-19-16908-t002:** Mean scores of nurses’ job satisfaction at the pre-pandemic moment and after the 3rd critical period of COVID-19.

	Pre-Pandemic Moment		After the 3rd Critical Period of COVID-19		
Dimensions/Scale	Mean	Standard Deviation	Mean	Standard Deviation	*p*-Value *
Satisfaction with the leadership	51.5	15.3	53.2	16.2	<0.001
Satisfaction with the organisation and resources	55.4	13.7	55.2	13.8	0.274
Satisfaction with professional recognition	54.2	16.7	58.3	17.8	<0.001
Satisfaction with the co-workers	59.4	14.4	61.9	14.6	<0.001
Satisfaction with recognition and remuneration	23.7	16.1	21.3	16.5	<0.001
Satisfaction with staffing	47.9	22.9	42.5	21.9	<0.001
Global Scale	49.8	12.2	50.3	12.7	<0.001

* Significance—Wilcoxon’s test.

**Table 3 ijerph-19-16908-t003:** Correlation between components of professional practice environments and nurses’ job satisfaction.

	Dimensions *	Dim 1	Dim 2	Dim 3	Dim 4	Dim 5	Dim 6	Global Scale
Components								
Structure								
Pre-pandemic	Cor †	0.599	0.510	0.411	0.383	0.361	0.360	0.610
	*p*-value ‡	<0.001	<0.001	<0.001	<0.001	<0.001	<0.001	<0.001
After the 3rd critical COVID-19 period	Cor †	0.663	0.629	0.421	0.536	0.307	0.533	0.710
	*p*-value ‡	<0.001	<0.001	<0.001	<0.001	<0.001	<0.001	<0.001
Process								
Pre-pandemic	Cor †	0.440	0.526	0.355	0.463	0.232	0.346	0.502
	*p*-value ‡	<0.001	<0.001	<0.001	<0.001	<0.001	<0.001	<0.001
After the 3rd critical COVID-19 period	Cor ‡	0.475	0.529	0.316	0.522	0.240	0.386	0.524
	*p*-value ‡	<0.001	<0.001	<0.001	<0.001	<0.001	<0.001	<0.001
Outcome								
Pre-pandemic	Cor †	0.544	0.507	0.337	0.469	0.468	0.360	0.606
	*p*-value ‡	<0.001	<0.001	<0.001	<0.001	<0.001	<0.001	<0.001
After the 3rd critical COVID-19 period	Cor †	0.530	0.563	0.407	0.507	0.349	0.441	0.610
	*p*-value ‡	<0.001	<0.001	<0.001	<0.001	<0.001	<0.001	<0.001

* Subtitles of the dimensions: Dim 1—Satisfaction with the leadership; Dim 2—Satisfaction with the organisation and resources; Dim 3—Satisfaction with professional recognition; Dim 4—Satisfaction with the co-workers; Dim 5—Satisfaction with recognition and remuneration; Dim 6—Satisfaction with staffing. † Cor—Correlation; ‡ Significance—Spearman’s correlation.

## Data Availability

The data that support the findings of this study are available from the corresponding author upon reasonable request.

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
