# Peer review of "Impact of COVID-19 on the Environments of Professional Nursing Practice and Nurses’ Job Satisfaction"

_ijerph, 2022, doi:10.3390/ijerph192416908_

Round 1
Reviewer 1 Report
Olga Maria Pimenta Lopes Ribeiro et al. submitted to IJERPH an article focusing to the impact of COVID-19 on the environments of professional nursing practice, assessing the nurses’ job satisfaction too.
This manuscript appears interesting for experts in this field and, if suitably modified and expanded, it could provide useful indications for future developments of healthcare workers.
Unfortunately, this version intended for refereeing does not show the progressive number of the "lines" on the side margins, it follows that it is not always easy to identify exactly the points intended for comments/suggestions.
Here are my suggestions:
- Introduction: "[...] requiring significant changes and adjustments in the environments of practice [4], especially in its early stage, in hospitals": please also consider the essential territorial aspects in which the Prevention Departments have been involved in the pandemic era, evaluating the following citation: DOI: 10.3390/healthcare10101906
- taking into account the denominator equal to 713 enrollable professionals, using the Sample Size Calculator, the enrolled population of 416 registered nurses satisfies the inferential statistical requirement, fine.
- was a preliminary pilot study carried out, useful for evaluating the intelligibility of the 37 items? If yes, please provide details in the manuscript;
- it would be important to highlight, in the discussions, whether different healthcare workers have been recruited in the pandemic context than the “eligible” ones, promoting the skills mix and any complementary training. Has this happened in your Organization, or did you have to obtain results with human resources available in the pre-COVID era only?
- focusing to the discussions paragraph, in specifically considering the nursing satisfaction level in the COVID era, please evaluate the opportunity to consider the contents described in the following articles: DOI: 10.3390/ijerph18041552 - DOI: 10.1016/j.apnr.2021.151416
- according to the Authors, what are the useful strategies to improve the professional satisfaction of registered nurses? What are the opportunities and future prospects that can contribute to the improvement and enhancement of healthcare workers? It would be useful to discuss these aspects, to support and validate your work.
The English language is easily understandable and the manuscript appears written with a good scientific soundness and with the logical rigor, typical of international publications.
Thank you for your efforts in perfecting this important article.
Author Response
Dear Reviewer,
We would like to thank you for your attentive reading of our manuscript and for the opportunity to improve our work. Your comments and suggestions were very helpful. Please find below the details of our revisions, as well as some clarifications. We hope they are clear and informative. Please note that the changes are highlighted in yellow in the main document.
We remain at your disposal should you have any questions.
Yours sincerely,
The co-authors

Reviewer 2 Report
Introduction
Please expand the introduction with literature on working conditions , burnout, physical labor, stress and also economic conditions among nurses. This is a very widely discussed topic and you provide barely a few items in your
manuscript.
Material and Methods
-Please be sure to add in the methodology a graph showing the research process, sample selection including inclusion and exclusion criteria (procedure paragraph)
- Were any questionnaires rejected from the analysis? Based on what criteria? Unfortunately, this is not specified in the paper.
-Were the subjects informed about the purpose of the study
- Please specify in the paper exactly when the material was collected-formulation before the Pandemic is insufficient.
- Was the exact same group of people surveyed in two time frames.Was the size the same in both cases?
Results
-Please underline and highlight the most relevant results
- part of the results please present in graphic form which will certainly be more interesting for the reader
Discussion
- please expand the discussion with the mentioned literature in the introduction (those to be added) and refer to your results in it
Limitations
Were there any limitations in the paper?Please add this paragraph because there certainly were and they are not placed in the paper
Author Response

(The authors gave the same response as above.)

Reviewer 3 Report
Dear Author,
the research is great! I found only some minor issues! You can find it attached!
All the best!

Author Response

(The authors gave the same response as above.)

Round 2
Reviewer 1 Report
The manuscript has all the characteristics to be favorably evaluated for its acceptance. The Authors have adequately complied with all the requests formulated, thank you.
Reviewer 2 Report
dear Authors
you can see the great work done in improving the quality of the manuscript
thank you for taking my comments into consideration
recommends the manuscript for publication